Legionella shows a diverse secondary metabolism dependent on a broad spectrum Sfp-type phosphopantetheinyl transferase

Tobias Nicholas J. 1
Ahrendt Tilman 1
Schell Ursula 2
Miltenberger Melissa 1
Hilbi Hubert 2 3
Bode Helge B. 1 4 h.bode@bio.uni-frankfurt.de
1 Fachbereich Biowissenschaften, Merck Stiftungsprofessur für Molekulare Biotechnologie, Goethe Universität , Frankfurt am Main , Germany
2 Max von Pettenkofer Institute, Ludwig-Maximilians-Universität München , Munich , Germany
3 Institute of Medical Microbiology, University of Zürich , Zürich , Switzerland
4 Buchmann Institute for Molecular Life Sciences, Goethe Universität , Frankfurt am Main , Germany
Silva Pedro
Electronic publication date: 2016 Nov 24
Publication date: 2016
Volume: 4
Electronic Location ID: e2720
Received 2016 Jun 29; Accepted 2016 Oct 25
Copyright: © 2016 Tobias et al.
Copyright year: 2016
Copyright holder: Tobias et al.
License: This is an open access article distributed under the terms of the Creative Commons Attribution License, which permits unrestricted use, distribution, reproduction and adaptation in any medium and for any purpose provided that it is properly attributed. For attribution, the original author(s), title, publication source (PeerJ) and either DOI or URL of the article must be cited.
License URL: https://creativecommons.org/licenses/by/4.0/

Keywords: Secondary metabolism, Legionella, Non-ribosomal peptide synthetase, Polyketide synthase, Natural products

Funding: Deutsche Forschungsgemeinschaft (DFG) Work in the Bode lab was supported by the Deutsche Forschungsgemeinschaft (DFG). The funders had no role in study design, data collection and analysis, decision to publish, or preparation of the manuscript.

==============================
Several members of the genus Legionella cause Legionnaires’ disease, a potentially debilitating form of pneumonia. Studies frequently focus on the abundant number of virulence factors present in this genus. However, what is often overlooked is the role of secondary metabolites from Legionella. Following whole genome sequencing, we assembled and annotated the Legionella parisiensis DSM 19216 genome. Together with 14 other members of the Legionella, we performed comparative genomics and analysed the secondary metabolite potential of each strain. We found that Legionella contains a huge variety of biosynthetic gene clusters (BGCs) that are potentially making a significant number of novel natural products with undefined function. Surprisingly, only a single Sfp-like phosphopantetheinyl transferase is found in all Legionella strains analyzed that might be responsible for the activation of all carrier proteins in primary (fatty acid biosynthesis) and secondary metabolism (polyketide and non-ribosomal peptide synthesis). Using conserved active site motifs, we predict some novel compounds that are probably involved in cell-cell communication, differing to known communication systems. We identify several gene clusters, which may represent novel signaling mechanisms and demonstrate the natural product potential of Legionella.

Introduction

The genus of Legionella is relatively diverse with 58 member species, 29 of which are known to cause disease in humans (Cunha, Burillo & Bouza, 2016). Legionellosis, infection with a member of the genus, can result in a form of pneumonia known as Legionnaires’ disease or the less severe, flu-like disease known as Pontiac fever. The first Legionella was identified following an outbreak of Legionnaires’ disease in 1976, and named Legionella pneumophila (Fraser et al., 2010). This species is responsible for a large proportion of Legionnaires’ cases, can often require hospitalization and is particularly dangerous for immuno-compromised patients (Schlossberg & Bonoan, 1998).

All Legionella spp. have a common association with water sources, surviving within amoebae, protozoa or slime moulds (Fields, Benson & Besser, 2002). Their association within microbial biofilm communities is also beneficial for their ability to survive and cause disease (Chaabna et al., 2013; Khweek et al., 2013). This close association between bacteria and protozoan host has led to a number of horizontal gene transfer events, significantly contributing to the intracellular fitness of these bacteria (Chien et al., 2004; Cazalet et al., 2004; Gimenez et al., 2011). Disease outbreaks often occur following contamination of industrial systems that help to spread the bacteria as infectious aerosols (Fraser, 1980; Nguyen et al., 2006). Following phagocytosis by eukaryotic cells, the bacteria are able to survive intracellularly, which is essential for disease progression.

Secondary metabolites are often small chemical compounds produced by a biosynthetic gene cluster (BGC), often consisting of either polyketide synthases (PKS) or non-ribosomal peptide synthetases (NRPS). These compounds are often not essential for survival but might have significant roles in niche adaptation and virulence. Briefly, PKS and NRPS are multifunctional enzymes that catalyze the condensation of carboxylic acid (PKS) (Hertweck, 2009) or amino acid (NRPS) building blocks (Sieber & Marahiel, 2005). PKS catalyze the formation of C-C bonds via the condensation of malonyl and acyl subunits that are enzyme bound, as in the case of type I PKS, which show similar protein domain architecture to eukaryotic fatty acid synthases (FAS). The catalytic functions of PKS and NRPS are organized in modules, with each module responsible for the incorporation and processing of one individual building block (different acyl or malonyl units for PKS or amino acids for NRPS). Due to these similar biochemical principles, hybrids of PKS and NRPS are also possible (Du & Shen, 2001). The biosynthesis of PKS and NRPS derived natural products as well as fatty acids requires specialized phosphopantetheinyl transferases (PPTases) that catalyze the post-translational transfer of the 4′-phosphopantetheinyl group from coenzyme A (CoA) to acyl (acyl carrier protein (ACP)) or peptidyl (peptidyl carrier protein (PCP)) carrier proteins also called thiolation (T) domains. These are components of the enzyme complexes of FAS, PKS and NRPS (Mootz, Finking & Marahiel, 2001; Mofid, Finking & Marahiel, 2002) and covalently link the biosynthesis intermediates to the enzyme complexes. PPTases in bacteria are classified as acyl carrier protein synthase (AcpS) or Sfp (required for surfactin production in Bacillus subtilis) enzymes and exhibit different substrate specificities. Sfp-PPTases are monomeric enzymes of approximately 240 aa (Mofid, Finking & Marahiel, 2002) that were shown to activate all kinds of T domains from FAS, PKS and NRPS by attachment of a phosphopantetheinyl group. AcpS PPTases on the other hand, are only half the size and are only functional for ACPs from FAS and type II PKS (Gehring et al., 1997; Mootz, Finking & Marahiel, 2001; Mofid, Finking & Marahiel, 2002). Therefore, most bacteria (especially those producing secondary metabolites) have two or more PPTases encoded in their genome.

The presence of PKS and NRPS is well established in all types of bacteria, for example, Streptomyces, Mycobacteria, Myxobacteria, Pseudomonas and Bacillus. Often these products are essential in a particular facet of their respective lifecycles. From Legionella, only four BGCs have been explored in detail with three secondary metabolites structurally elucidated to date (Fig. 1) (Ahrendt et al., 2013; Shevchuk et al., 2014; Burnside et al., 2015; Johnston et al., 2016a). Legioliulin (1), a product of a trans-AT PKS cluster first identified in L. dumoffii, was reported originally in 2004 (Amemura-Maekawa et al., 2004) and biological activity assays failed to determine a role for the compound beyond fluorescence (Ahrendt et al., 2013). This study investigated the possibility that legioliulin is required for intracellular survival and ultimately failed to assign a biological function. On the other hand, a transposon mutagenesis library of L. pneumophila revealed a polyketide that interferes with lysosomal degradation during infection of both protozoa and macrophages (Shevchuk et al., 2014). Legiobactin (2) is a siderophore involved in iron sequestration (Cianciotto, 2007) and the unusual polyketide legionellol A (3) is involved in sliding motility and might additionally act as a surfactant (Johnston et al., 2016a). Despite all Legionella strains containing several BGCs, no further research has explored the roles of their respective products. To attempt to further explore the possibility that secondary metabolites are an important part of the Legionella lifecycle, we performed genome-wide comparisons of 15 genome sequences from Legionella, paying particular attention to the prevalence of BGCs. We explore possible structures and functions for these BGCs.

Figure 1 Structures of the known Legionella natural products legioliulin, legionellol and legiobactin as well as PPTase inhibitor used in this study (4–6).

Materials and Methods

Culture conditions and DNA methods

The Legionella strains L. pneumophila JR32 (Sadosky, Wiater & Shuman, 1993) and ΔicmT (Segal & Shuman, 1998), L. longbeachae NSW150 and L. parisiensis DSM 19216 were grown in N-(2-acetamido)-2-aminoethanesulphonic acid (ACES) yeast extract (AYE) broth (Feeley et al., 1979) or on buffered charcoal yeast extract agar (Difco, Detroit, MI, USA) for three days at 37 °C. E. coli BL21 Star (DE3) (Novagen) was cultured in LB medium supplemented with 40 μg/mL kanamycin (Kan) and 100 μg/mL Ampicillin (Amp) (Carl Roth, Karlsruhe, Germany), if necessary. Cells were harvested and DNA was extracted using the Puregene Yeast/Bacteria Kit B (Qiagen) according to the manufacturer’s recommendations.

Genome sequencing, assembly and annotation

Shotgun sequencing of Legionella parisiensis DSM 19216 was performed using a Genome Sequencer FLX (Roche) by MWG Genomics (Munich). Assembly was performed using the Celera Assembler (v5.3) and quality assessed using QUAST (Gurevich et al., 2013). Sequencing yielded a total of 290,164 reads with average read length of 353 bp. The L. parisiensis genome was assembled into a total of 226 contigs (115 ≥ 1 kb) with an N50 of 65,672 bp and a predicted genomic coverage of 25. Genome annotation was performed using prokka (v1.12) (Seemann, 2014). Abricate was used to identify common antibiotic resistances (https://github.com/tseemann/abricate).

Phylogenetic analysis

Fourteen Legionella genomes were downloaded from NCBI (Table 1), their protein fasta files extracted and together with L. parisiensis, ortholog families were identified using proteinortho5 (Lechner et al., 2011). Protein singletons identified in only a single species were removed from further analysis. The presence or absence of all ortholog families were used to generate a gene content tree using the binary function associated with RAxML (Stamatakis, 2014) and the gamma model of rate heterogeneity and a random number seed for parsimony inferences. Protein sequences of the ACPs of E. coli (ACPS, WP_000986025.1) and Bacillus subtilis (Sfp-like, WP_003234549.1) were taken from the NCBI website and used to identify homologs in each species with Blastp. Protein sequences were aligned using ClustalW, and phylogenetic trees of the PPTases were created using the PhyML plugin attached to Geneious (v6.1.6) (Guindon et al., 2010). Branch formation was supported with bootstrapping (n = 1,000).

Table 1 All genome details for Legionella spp. used in this study.

Species	Genome accession no.	Source	Reference	
Legionella anisa str. Linanisette	NZ_CANP00000000.1	Clinical sample	Pagnier et al. (2014)	
Legionella cherrii DSM19213	NZ_JHYM00000000.1	Thermally altered water	Brenner et al. (1985)	
Legionella drancourtii LLAP12	NZ_ACUL00000000.2	Environmental water source	Gimenez et al. (2011)	
Legionella fairfieldensis ATCC49588	NZ_JHYC00000000.1	Cooling tower	Thacker et al. (1991)	
Legionella geestiana DSM21217	NZ_JHYN00000000.1	Domestic hot water	Dennis et al. (1993)	
Legionella lansingensis DSM19556	NZ_JHWF00000000.1	Clinical sample	Thacker et al. (1992)	
Legionella longbeachae NSW150	NC_013861.1, NC_014544.1	Clinical sample	Cazalet et al. (2010)	
Legionella moravica DSM19234	NZ_AUHS00000000.1	Cooling tower	Wilkinson et al. (1988)	
Legionella norrlandica strain LEGN	NZ_JNCF00000000.1	Biopurification system of wood processing plant	Rizzardi et al. (2015)	
Legionella oakridgensis ATCC33761	NZ_CP004006.1, NZ_CP004007.1	Cooling tower	Brzuszkiewicz et al. (2013)	
Legionella pneumophila subsp. pneumophila str. Philadelphia 1	NC_002942.5	Clinical sample	Chien et al. (2004)	
Legionella sainthelensis ATCC35248	NZ_JHXP00000000.1	Surface water	Campbell et al. (1984)	
Legionella shakespearei DSM23087	NZ_AREN00000000.1	Cooling tower	Verma et al. (1992)	
Legionella wadsworthii DSM21896	NZ_JNIA00000000.1	Clinical isolate	Edelstein et al. (1982)	
Legionella parisiensis DSM19216	LSOG00000000	Cooling tower	This study	

Cloning and expression of LparPPTase

pCOLA_Duet1 (Novagen) was used as a vector for overproduction of the PPTase from L. parisiensis. The PPTase gene was amplified using primers Lpar_PPtase_Fw_SacI (GAGCTCGATGATCATTACCGAATTTAACCCT) and Lpar_PPtase_Rv_PstI (GTTCTGAATTAGGGGCAACGTGTCGAC) (synthesized by Sigma-Aldrich, St. Louis, MI, USA). Both the PCR product obtained and pCOLA_Duet1 were digested with SacI and PstI (Fermentas). Digestion products were separated by gel electrophoresis and desired fragments isolated with Gene JETGel extraction kit (Fermentas). Isolated fragments were ligated for 1 h at room temperature using T4-ligase (Fermentas). After ligation, E. coli DH10B was transformed with the ligation mixtures in a 1 mm cuvette by electroporation at 1,250 V, 200 Ω and 25 μF. Cells were plated on LB-Kan agar and incubated overnight at 37 °C. Colonies were picked and inoculated in LB-Kan media for plasmid extraction. The plasmids obtained were sequenced, and pCOLA_LparPPTase plasmids transferred into E. coli BL21 Star. Positive colonies were picked and cells were transformed with pUC18_indC (Brachmann et al., 2012) and grown on LB-Kan-Amp agar. Cells were grown to an OD600 of 0.5 at 37 °C at which time cultures were induced with 0.1 mM isopropyl-β-D-thiogalactopyranoside (IPTG) (Fermentas), and the cultures were incubated at 16 °C overnight. Following induction of LparPPTase in pUC18_indC, cells were pelleted and resuspended in deionized water for easy visualization of the blue pigment produced by IndC.

Legionella in vivo inhibition

L. parisiensis was grown to an OD600 of 0.1 in 200 μl AYE broth in a 96-well plate at 37 °C. Putative PPTase inhibitors 4–6 (Foley et al., 2014) were then added in different concentrations, and the cells were allowed to grow for 24 h. For visualization of legioliulin production in L. parisiensis, the cells were illuminated with long-wave UV-light. The MIC of the PPTase inhibitors were tested in triplicate on L. parisiensis, L. pneumophila and L. longbeachae using the OD600 value.

Secondary metabolite identification

Secondary metabolites were identified using antiSMASH 3.0 (Weber et al., 2015) with the optional ClusterFinder algorithm activated. The results from each genome were then aligned using Mauve (Darling et al., 2004), a BLAST based analysis program, to identify homologous clusters. Using this method, we assembled some clusters that were split across different contigs by sequence similarity, additionally taking into account the predicted substrate specificities and domain modifications from each unassembled module. The sequence for the isocyanide synthase cluster, isnAB, was taken from Xenorhabdus nematophila (Crawford et al., 2012) and identified in Legionella species using BLASTp (v2.2.29) as a part of the BLAST+ suite (Camacho et al., 2009).

Results

Genome of L. parisiensis

Purified genomic DNA from L. parisiensis DSM 19216 was used for shotgun sequencing. Assembly using Celera (v5.3) revealed a 4,232,283 bp genome with a GC content of 37.98% and was predicted to contain 3,916 protein-coding sequences (CDS). This Whole Genome Shotgun project has been deposited at GenBank under the accession number LSOG00000000. The version described in this paper is version LSOG01000000.

Genome wide analyses

Together with the 14 other Legionella genomes (Table 1) we identified all protein ortholog families in Legionella (Table S1). The core genome of the 15 Legionella species consists of 711 coding sequences and includes a type II secretion system as well as the Dot/Icm system. The conserved type II secretion system is essential for intracellular survival and growth (Hales & Shuman, 1999; Polesky et al., 2001; Rossier, Starkenburg & Cianciotto, 2004) as well as promoting growth at low temperatures (Söderberg, Rossier & Cianciotto, 2004). The Dot/Icm system is already known to be ubiquitous in all strains (Feldman et al., 2005). The effectors secreted by this system work in concert to evade the phagosome and form the Legionella-containing vacuole allowing the bacteria to grow intracellularly (Isberg, O’Connor & Heidtman, 2009; Ensminger, 2016).

Using the program abricate, we additionally analysed the genomes for possible antibiotic resistance genes. In L. anisa, L. cherrii, L. longbeachae, L. sainthelensis and L. wadsworthii beta-lactamase resistance was identified with no other antibiotic resistance genes present. However, several multi drug efflux pumps were also found in the genomes (Table S2). Using the amino acid sequences of all annotated coding sequences from each strain, we determined ortholog families using proteinortho5 (Lechner et al., 2011). From these ortholog families, we produced a phylogeny representing the gene content based upon the presence or absence of each protein ortholog family. Following analysis of all Legionella strains and their BGCs, we constructed a map of each BGC common to more than a single species based on the protein sequence identity (Figs. 2 and 3). Bacteriocins are a class of ribosomally synthesized peptides with antibacterial properties. They are classified based on their mode of action and size (Yang et al., 2014) and are typically used to attack other bacteria competing in similar environments (or sometimes have broad-spectrum activity) but contain resistance mechanisms to avoid self-harm (Cotter, Hill & Ross, 2005). This analysis revealed that there is a range of different bacteriocins present in Legionella with all species containing at least one cluster with L. geestiana, L. oakridgensis and L. shakespearei containing bacteriocins not present in any other species (Figs. 2 and S1; Table S3).

Figure 2 Legionella phylogeny based on presence or absence of ortholog families together with a summary of orthologous BGCs found in two or more Legionella species.

BGCs were identified using antiSMASH (Weber et al., 2015) and nucleotide sequences were aligned using Mauve (Darling et al., 2004) to determine those that were similar. Ortholog presence was first determined using proteinortho5 (Lechner et al., 2011). The gene content tree was then constructed using RAxML, based on the presence or absence of each ortholog. BGCs are separated according to the class of compound produced. Cluster letters refer to those genetic schematics shown in Fig. 3 and compound numbers refer to those found in Figs. 1 and 6. A full list of BGCs can be found in Table S3. The gene cluster encoding IsnAB is not detected by antiSMASH but is a known BGC responsible for the biosynthesis of isonitrile containing compounds that are widespread in bacteria (Brady et al., 2007).

Figure 3 Representative examples of BGCs found in multiple Legionella species as identified in Fig. 2.

Protein domain architecture as determined from NCBI’s conserved domain database for NRPS (green) and PKS (red) encoding genes are also shown. Each circle represents an individual domain of the respective PKS or NRPS (domains not to scale). The PKS from L. pneumophila (I) contains a C-terminal condensation domain typical of those seen in NRPSs, which is also capable of polyketide chain release. Clusters O-U can be found in Fig. S1. All clusters are in Table S3.

Only a single PPTase was identified in Legionella, which activates natural product biosynthesis clusters in vitro

Interestingly, the ortholog analysis identified only a single Sfp-like PPTase in all of the analyzed Legionella genomes (Fig. 4). No AcpS-like PPTase that is usually involved in fatty acid biosynthesis exists (Mofid, Finking & Marahiel, 2002). PPTases are required to post-translationally attach a 4′-phosphopantetheine arm from CoA to the serine residue contained in the T (ACP or PCP) domain and therefore are essential for fatty acid, polyketide and non-ribosomal peptide biosynthesis (Walsh et al., 1997; Stack, Neville & Doyle, 2007).

Figure 4 Maximum likelihood phylogeny created using PhyML of PPTases identified in Legionella genomes and their relationship to a selection of PPTases from other bacteria.

Scale represents amino acid substitutions per amino acid position. Bootstrapping (n = 1,000) was used to support branch formation.

Unsurprisingly, within this group, Legionella PPTases form a distinct branch (Fig. 4). To test if the L. parisiensis Sfp-like PPTase could activate a NRPS, IndC from Photorhabdus luminescens (Brachmann et al., 2012) and the PPTase from L. parisiensis were co-produced in E. coli BL21 Star. IndC produces the blue pigment indigoidine by condensation of two glutamines. While indC is constitutively expressed in this experiment, the L. parisiensis PPTase gene expression was under control of an IPTG-inducible promoter. Addition of IPTG and consequent PPTase expression led to blue pigment production (Fig. S2A). As the E. coli Sfp-type PPTase, EntD, is not capable of activating IndC (Brachmann et al., 2012), any production of indigoidine must be activated by the PPTase from L. parisiensis. Harvesting and resuspension of the colored cells in water shows a bright blue pigmentation of the IPTG-induced culture (Fig. S2B).

Inhibition of legioliulin production and growth in L. parisiensis

To rule out the possibility that any PPTase was missed in this analysis, we used 2-pyridinyl-N-(4-aryl)-piperazine-1-carbothioamides (4–6), specific inhibitors of bacterial Sfp-like PPTases (Foley et al., 2014), to shut off legioliulin production. Legioliulin production and growth are closely linked. Bacterial growth was measured at OD600, and legioliulin production was observed under long-wave UV-light. The addition of 1 μg/mL of inhibitor 4 resulted in a total loss of legioliulin production. We then used different concentrations of 4 to determine substance effectivity (Fig. 5). Concentrations as low as 0.4 μg/mL of 4 showed an inhibition in legioliulin biosynthesis and growth. For compounds 5 and 6, initial inhibitory effects were observed at concentrations of 0.75 and 6 μg/mL, respectively. Similarly, growth inhibition was observed for L. pneumophila and L. longbeachae (Table S3).

Figure 5 Inhibition of legioliulin production resulting in fluorescence (at 366 nm) in L. parisiensis by PPTase inhibitors 4–6.

Biosynthetic gene clusters

During the secondary metabolite analysis, we used antiSMASH to predict BGCs and extracted all those containing predicted siderophore, PKS, NRPS, lantipeptide or bacteriocin clusters. With the optional ClusterFinder algorithm activated (Cimermancic et al., 2014), we also examined all putative and saccharide-like clusters for misclassification (Table S3). Strains contained between 15 and 36 BGCs in total with NRPS clusters being the most prevalent. The most widespread PKS, NRPS and siderophore clusters found in Legionella are shown in Fig. 3 highlighting the overall synteny as well as the domain architecture of the natural product synthases.

Non-ribosomal peptide synthetase product predictions in Legionella

PKS and NRPS specificity can often be predicted based upon the DNA sequence and comparisons to experimentally validated studies (Stachelhaus, Mootz & Marahiel, 1999; Challis, Ravel & Townsend, 2000; Yadav, Gokhale & Mohanty, 2003). In the case of the Stachelhaus code, conserved motifs in the adenylation (A) domain are used to predict substrate specificity. These conserved motifs and their respective specificities were confirmed by single nucleotide mutations resulting in either a loss of, or relaxation of substrate specificity (Stachelhaus, Mootz & Marahiel, 1999). Prediction using a hidden Markov model based approach is also available to predict specificities of either A domains from NRPS or acyltransferase (AT) domains from PKS and is integrated into antiSMASH (Minowa, Araki & Kanehisa, 2007; Weber et al., 2015). NRPSPredictor2, unlike the previous two methods, uses a support vector machine to predict specificities (Röttig et al., 2011). These methods formed the basis to predict the structures of the natural products produced by BGCs in Legionella. In many Legionella BGCs, the specificities for A domains involved in the activation of the correct amino acid in the NRPS could either not be predicted or showed variable results when these different algorithms were used. We therefore only attempted to predict resulting structures where a consensus among the three methods was reached. Several low molecular weight natural products produced from monomodular NRPS could be predicted assuming non-iterative use of these NRPS modules (Fig. 6).

Figure 6 Theoretical structures of compounds 7–10 predicted from the clusters B–D, G and I (monomodular NRPS), shown in Fig. 2.

Monomodular NRPS are predicted to produce modified amino acids or dipeptide derivatives that have also been identified in different fungi (Forseth et al., 2013). In a relatively rarely described phenomenon, NRPS domains may be re-used during product biosynthesis resulting in peptides longer than expected from the NRPS domain architecture. An example of such an iterative use is due to the action of the thioesterase domain which, following a single round of biosynthesis, must oligomerize the enzyme bound peptide product before release from the NRPS (Shaw-Reid et al., 1999; Bruner et al., 2002; Hoyer, Mahlert & Marahiel, 2007; Felnagle et al., 2008). Due to the relative infrequency that this happens, we assumed non-iterative use of domains for all structural predictions. Cluster B (Fig. 3) encodes a NRPS/PKS hybrid that is suggested to produce a valine elongated by a single polyketide elongation using malonyl-CoA with the resulting product, dependent on the thioesterase (TE) function, might be linear (7a) or cyclic (7b). Cluster C encodes a monomodular NRPS that is predicted to produce a N-formylated amino acid that is either reduced by the C-terminal reduction (Red) domain to the aldehyde (8a), or the alcohol (8b) that can then by cyclized non-enzymatically to form an oxazoline ring (8c). The acylated amino acid derived from cluster D can undergo similar transformation resulting in structurally related compounds (9a, 9b, 9c). Cluster G and I are very similar to C but the NRPS is terminated by a TE domain resulting again in either a linear (10a) or cyclic product (10b).

Discussion

Secondary metabolism in Legionella is under-pinned by a broad spectrum PPTase

Following the sequencing of the L. parisiensis genome, we noted the presence of 32 BGCs, as predicted by antiSMASH (Table S3). We then further investigated a selection of other Legionella strains to obtain a snapshot of the secondary metabolite potential of the genus. Through ortholog clustering we looked specifically for genes that are known to be essential in secondary metabolism.

Interestingly, this diversity in secondary metabolites gene clusters appeared to be controlled by a single Sfp-like PPTase in all Legionella strains analyzed, L.Ppt (Legionella PPTase, Fig. 4). This PPTase may therefore be capable of activating all different carrier proteins involved in polyketide and non-ribosomal peptide biosynthesis as well as fatty acid biosynthesis, a part of the primary metabolism as has been seen before (Losick & Isberg, 2006). A precedent for this has been made in Pseudomonas aeruginosa, which carries only a single broad spectrum PPTase that is active in both primary and secondary metabolism (Seidle, Couch & Parry, 2006). However, the veracity of this hypothesis is yet to be definitively determined in Legionella. Following identification of only one PPTase, we used an indigoidine production assay to confirm a role for L.Ppt from L. parisiensis in secondary metabolism. The enzyme was able to activate the NRPS IndC from P. luminescens, even though no NRPS product is known for any Legionella strain so far confirming this function. To investigate the effect on suppression of the Sfp-type PPTase, we grew L. parisiensis in the presence of Sfp-type PPTase inhibitors (Foley et al., 2014) and showed that legioliulin production, in addition to cell viability, is halted (Table S4; Fig. 5). The importance of this is that if only a single PPTase controls both primary and secondary metabolism, PPTase inhibitors may be effective as monotherapeutic drugs with multi-target effects (Silver, 2007) resulting from the loss of several functional ACP or PCP proteins, inhibiting essential fatty acid and secondary metabolite biosynthesis. Although fatty acid biosynthesis has been questioned as a general target for antibiotic therapy (Parsons & Rock, 2011), the parallel inhibition of fatty acid, virulence factor and signaling compound biosynthesis might make PPTase inhibitors powerful antibiotics or drugs that could also work against intracellular pathogens, where fatty acid biosynthesis is essential (Yao et al., 2014).

Reconstruction of BGCs and ortholog clustering highlight the diversity of potential secondary metabolites in Legionella

Only the structures of legioliulin (1), legiobactin (2) and legionellol (3) have been solved (Fig. 1) while one other PKS derived compound has been implicated in lysosomal degradation (Shevchuk et al., 2014). Legioliulin is a trans-AT PKS derived fluorophore (Fig. 3A). However, beyond fluorescence of bacterial strains containing the gene cluster, a biological function was not defined for legioliulin. This is perhaps unsurprising given that only the strains amoebic intracellular growth capabilities were tested while the species has been isolated from both environmental and clinical sources in both fluorescing and non-fluorescing forms (Igel, Helbig & Lück, 2004). The cluster of coding sequences responsible for legionellol, a hydrophilic molecule involved in lipid scaffolding, has been ascribed to a number of small discrete genes (lpg2223-41) coding for different domains in L. pneumophila (Fig. S3) (Johnston et al., 2016b).

One disadvantage with short read sequencing technologies is that long gene sequences that are prone to containing repetitive sequences may not be properly assembled. This may be the case for the PKS and NRPS gene clusters that we have examined here, as some are known to be highly repetitive such as the mycolactone PKS (Stinear et al., 2004) or the syringopeptin NRPS (Scholz-Schroeder, Soule & Gross, 2003). Although Legionella probably do not contain examples as extreme as mycolactone or syringopeptin, it is possible that the misclassified saccharide-like clusters or some of the contigs containing clusters at their respective termini are in fact collapsed BGCs due to poor assembly. Despite this, we found significant conservation of some BGCs, although this was not always reflected in the phylogenetic tree composed of all coding sequences. For example, Cluster F (Fig. 2) is present in species that appear more dissimilar with respect to their gene content. This observation may be in part explained by the amount of horizontal gene transfer that is reported to occur in this genus leading to a greater diversity of coding sequences (Gomez-Valero et al., 2011).

Cluster F, the most prevalent cluster, is a NRPS consisting of a single module containing an A, T and C domain, however it was not limited to a given clade of bacteria suggesting it is probably either dispensable for growth and survival, or it plays a more general role. Perhaps more interesting are the clusters that are exclusive to certain clades such as clusters E and K, a NRPS and type III PKS, respectively as well as clusters M and N, which are both siderophores. The apparent maintenance of these clusters in specific clades may be representative of essential functions in their particular environment. However, experimental evidence is needed to verify the veracity of this hypothesis. Siderophores are a well-known virulence factor of many bacteria and the structure of legiobactin (2) has already been elucidated in L. pneumophila (Cluster M (Burnside et al., 2015)). It is reported as having an identical structure to rhizoferrin (Drechsel et al., 1991; Burnside et al., 2015) and is essential for ferric iron uptake during infection of the lungs (Liles, Scheel & Cianciotto, 2000; Robey & Cianciotto, 2002; Allard et al., 2009; Chatfield et al., 2012).

Several Legionella strains also encode homologs of isnA and isnB that have been shown to be involved in the biosynthesis of isonitrile containing natural products that are widespread among bacteria (Brady et al., 2007). Specifically, isnA and isnB encode proteins that, together, produce an inhibitor of insect phenoloxidase that has been shown to be important in defense against host immune responses in entomopathogenic bacteria (Brady et al., 2007; Crawford et al., 2012). In Legionella, a helix-turn-helix domain protein and a cytochrome P450 oxidase are always associated with the cluster (Fig. 3L). In Pseudomonas, the isnAB cluster is part of a larger BGC and does not make the phenoloxidase inhibitor. There, the IsnAB homologs PvcA and PvcB are encoded as a part of the pyoverdine BGC where they are involved in maturation of the siderophore pyoverdine (Drake & Gulick, 2008).

Analysis of clusters B–D reveals the presence of NRPS that are clustered with genes encoding a transcriptional regulator. Although not definitive, this provides evidence supporting a role for these products as novel signaling compounds as seen in other Gram-negative bacteria (Brachmann et al., 2013; Brameyer et al., 2015). If this is indeed the case, its significance lies in the fact that the bacteria occupy a relatively diverse environment and the signals may be specific for their respective niches.

In addition to the more conserved clusters found in several strains, unique clusters have been identified that are present only in individual strains (Fig. S3). Among them is another trans-AT PKS in L. cherii that might be responsible for the described red fluorescence of this strain that also gave it its name. However, the red fluorescence might also be derived from the legioliulin cluster, also encoded in this genome, when a starting unit other than cinnamic acid is used that could result in a red-shift of the resulting fluorophore. Different PKS/NRPS hybrids are encoded in L. anisa, L. parisiensis and L. longbeachae that additionally encode type I PKSs that could also be involved in the production of unusual fatty acids or lipids required for their particular niche (Fig. S3).

There are a large number of diverse and interesting BGCs in Legionella that have thus far been unexplored. Although few are conserved across species, we cannot rule out the possibility that these BGCs are providing important chemical compounds to their respective strains, whether for signaling, or otherwise. The lack of cluster conservation further reinforces the notion that this genus is a large, untapped reservoir for novel secondary metabolite discovery. Given the association of these bacteria with protozoa in the environment and the interaction of the pathogenic strains with human phagocytic cells, bioactive metabolites originating from this genus may have activity against eukaryotic targets making this an interesting area of future research.

Supplemental Information

Supplemental Information 1 All ortholog families.

All ortholog families for the 15 analysed Legionella strains. Shown are the protein accession numbers for species available in the NCBI database. For L. parisiensis, locus tags are indicated. Annotated functions in each species are also shown.

Click here for additional data file.

Supplemental Information 2 CDS mentioned in the text.

Coding sequences mentioned in the text and the respective orthologs in each Legionella strain, where present.

Click here for additional data file.

Supplemental Information 3 All BGCs.

All biosynthetic gene clusters as identified by antiSMASH.

Click here for additional data file.

Supplemental Information 4 MIC of PPTase inhibitors.

Minimal inhibitory concentration (MIC) [μg/ml] of compounds 4–6 against L. pneumophila, L. longbeachae and L. parisiensis. Mean values of triplicate experiments are shown.

Click here for additional data file.

Supplemental Information 5 Clusters O-U as specified in Fig. 2.

Representative examples of clusters O-U from Fig. 2. For a full list of clusters, refer to Table S3.

Click here for additional data file.

Supplemental Information 6 Indigoidine production in E. coli by PPTase from L. parisiensis.

pUC18_indC and pCOLA_LparPPTase were used to transform E. coli and subsequently induced with IPTG. Indigoidine production can be seen in (A) E. coli grown in LB medium and more clearly following (B) pelleting of the E. coli cultures and resuspension in deionized water.

Click here for additional data file.

Supplemental Information 7 Unique BGCs in Legionella.

Selection of unique clusters identified in Legionella genomes. The legionellol cluster is shown at the bottom. For a full list of clusters, see Table S3. The domain architecture shown (below PKS (red) and NRPS (green) coding sequences) was determined using NCBI’s conserved domain database.

Click here for additional data file.

The authors are grateful to David J. Maloney for samples of PPTase inhibitors 4–6.

List of Abbreviations

PKS polyketide synthase

NRPS non-ribosomal peptide synthetase

BGC biosynthetic gene cluster

AT acyltransferase

A adenylation

T thiolation

C condensation

TE thioesterase

Additional Information and Declarations

Competing Interests

Author Contributions

DNA Deposition

The authors declare that they have no competing interests.

Nicholas J. Tobias conceived and designed the experiments, performed the experiments, analyzed the data, wrote the paper, prepared figures and/or tables, reviewed drafts of the paper.

Tilman Ahrendt conceived and designed the experiments, performed the experiments, analyzed the data, wrote the paper, prepared figures and/or tables, reviewed drafts of the paper.

Ursula Schell conceived and designed the experiments, performed the experiments, analyzed the data, contributed reagents/materials/analysis tools, reviewed drafts of the paper.

Melissa Miltenberger performed the experiments, reviewed drafts of the paper.

Hubert Hilbi conceived and designed the experiments, performed the experiments, analyzed the data, contributed reagents/materials/analysis tools, reviewed drafts of the paper.

Helge B. Bode conceived and designed the experiments, analyzed the data, contributed reagents/materials/analysis tools, wrote the paper, prepared figures and/or tables, reviewed drafts of the paper.

The following information was supplied regarding the deposition of DNA sequences:

Genbank: LSOG00000000.

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
