# Peer review of "Legionella shows a diverse secondary metabolism dependent on a broad spectrum Sfp-type phosphopantetheinyl transferase"

_PeerJ, doi:10.7717/peerj.2720_

## Round 0.1 · original submission · Major Revisions

· Academic Editor

Major Revisions

Please address all concerns from the three reviewers, especially regarding the presentation of results and discussion.
Personal observations from the editor:

Please improve title/legend of Supporting Figure 1 to clarify the experimental conditions in each panel.

Figure 4 does not include the data form BGC O-U. I wonder if the data present in this figure might become more readable in some aleternative form, as the very small font size is likely to become virtually unreadable when the figure is shrunk to fit to the journal's page size.

I agree with reviewer 1 regarding the level of detail present in the introduction: I would not go so far as saying that "it reads like a review", but at several points it gives the impression that a massive amount of data is being presented without a very clear indication of how important those data will be for your investigation. Could some of that information be presented when assessing the results/discussion instead?

How similar is the Legionella Sfp to other Sfps? sequnce identity to other Sfp, domain organization, etc. might be interesting for enzymologists.

Reviewer 1 ·

Basic reporting

A comprehensive genomic and biochemo- informatics analysis and well written.


Abstract Secondary metabolites can be part of the virulence repertoire as shown and discussed later so why to state it “however, what is often overlooked “… ,

Introduction
Should be shortened to focus on the background that is pertinent only to this study the section on quorum sensing can be removed


Discussion
The description of the different categories of BGCs the main body of the MS should be in the results section with specific subheadings all under the major heading BGC as these are the products of the analysis, the significance and of each class and its broader implication should be in the discussion section.
This section should usually begin with a succinct summary of the main findings rather than a description of the limitations of the genomic analysis that was conducted here, this paragraph should be moved.

Experimental design

The authors could and should annotate that single Sfp like PPTase such as L(p)Pp etc
The finding of a single PPTase that activates secondary metabolism is supported by indigoidine production through activation by L.PpT . The finding that primary metabolism of fatty acid biosynthesis is indeed very significant if “drug target ability” is concerned which primarily is due to inhibition of fatty acid synthesis
For the latter the authors, potential “drug-ability” could be much strengthened by proof of essentiality through an allelic exchange experiment using an inducible extra copy of the legionella PPTase.
Alternatively they could easily prove that it activates the highly conserved and essential protein ACP of the FAS II system by using a highly purified Legionella ACP (avoiding background activity of E coli AcpS traces that can convert it just in excess of CoA ) with Legionalla PPTase, looking for gel shifted ACP following P-pant binding.

Validity of the findings

The findings are valid

Additional comments

I believe that the MS can be strengthen if the introduction and discussion can be shortened so that the manuscript will be focused on the results obtained and not be read like a review , this also applies to the extensive references that sum up to 116 !


The distinctive role of PPTases in either primary and secondary metabolism is well established in M. tuberculosis and where acid biosynthesis is a prove n drug target in mycobacteria . The studies in that organism are a good comparator for the classification, paradigm and role of two PPTase systems in a pathogenic intracellular bacteria. Moreover, the potential “drug target ability “ of mycobacterial Sfp like PPTase PptT was shown in vitro and in vivo , recently its role as an activator of ACP of the FAS II system was also shown.
The final summary P 17 lines 15-23 is too broad, basically there are conserved and non conserved PKSs and NRPs, so then the conclusion the first category are likely essential for all common house keeping functions of the genus and the non conserved are the more specified function that are species specific , and these are that confer the breadth of the repertoire of BGCs of legionella genus.

·

Basic reporting

This is a technically well-written paper, with interesting insights in cross-species conservation and differences in secondary metabolite clusters in Legionella.
It is missing the "Conclusions" header from the template, but given the nature and extend of the discussion, I believe that to be a valid choice.

An accession number for the newly sequenced genome is given, but seems to be embargoed by the NCBI, I could not access the L. parisiensis genome. All other genomes are clearly listed in table 1 and easy to find.

Experimental design

The methods are detailed, and the supplemental material contains more information that quickly allows to validate the findings.

Apart from some surprising mismatches in figure 2, a quick check with antiSMASH identifies a comparable set of clusters in the 14 available genomes. My antiSMASH runs nicely agree with the data provided by the authors in table S3.

The experiments to validate the Sfp-like phosphopantetheinyl transferase activity is following the usual protocol.

Validity of the findings

No Comments.

Additional comments

## Summary

In their paper "Legionella shows a diverse secondary metabolism dependent on a single Sfp-type phosphopantethinyl transferase",
Tobias et al. compare the predicted secondary metabolite biosynthesis potential of 15 Legionella species. While the 15 strains harbour
quite some diversity in their biosynthetic potential, all PKS and NRPS substrate activations seem to be catalyzed by a single Sfp-like
phosphopantethinyl transferase. All species also harbour one or more RIPP clusters, possibly for use in signalling systems.

This is a technically well-written paper, with interesting insights in cross-species conservation and differences in secondary metabolite clusters in Legionella.
The methods are detailed, and the supplemental material contains more information that quickly allows to validate the findings.

## Major concerns:

* Mismatch between observed clusters (my own antiSMASH runs and table S3) and clusters reported in Figure 2:
* NZ_CP004006 contains 3 clusters (ignoring ClusterFinder predictions): 1 siderophore, 1 bacteriocin, 1 NRPS-like cluster. The siderophore and bacteriocin clusters are missing from fig. 2.
* NZ_JHYN00000000 siderophore and bacteriocin clusters are missing from fig. 2.
* NZ_JHWF00000000 has 2 bacteriocin clusters, fig. 2 shows 1 cluster only
* NZ_JNCF00000000 has 2 bacteriocin clusters, fig. 2 shows 1 cluster only
* NZ_AREN00000000 misses 2 bacteriocin and 2 siderophore clusters in figure 2.
* NZ_JHXP00000000 misses 1 bacteriocin in figure 2.
* NZ_JHXP00000000 misses the siderophore cluster.
Notably, all those clusters can be found in table S3. If they are omitted from figure 2 on purpose, this need to be explained in the paper.
* I could not look at the L. parisensis described in this study, as the sequence is not yet available.

## Minor concerns

* Why does the study use other WGS genomes for the comparison but skip some available complete genomes from GenBank?
I understand why multiple L. pneumophilia strains would not help much, but L. fallonii (LN614827.1) might have been an interesting addition,
as well as the unclassified Legionella sp. CP011105.1 and LN681225.1
* I applaud the authors for the quality of their bioinformatics methods section. Still, especially with a 454 FLX assembly, the options passed to Celera Assembler
would be of interest.
* In line 239-241, the authors speculate about horizontal gene transfers. It would be interesting to see if there were any transposon scars around the clusters.
* It would be nice if "isnAB" would be explained in the figure 2 legend, as it's not a very common term.

## Typos and grammar
* Line 50-52: "The system works by... with well over 300 now identified and 5885 predicted." is a bit confusing to read.
* Line 192: I believe the more common term is "multi drug efflux pump", not "multi drug resistant efflux pump", but both can be found in the literature.

Reviewer 3 ·

Basic reporting

The manuscript from Tobias et al., describes comparative genome analysis of 15 strains from the genus Legionella, usually known for their pathogenicity to humans. Special emphasis was made on the analysis of their secondary metabolites and a large variety of biosynthetic gene clusters was detected using the online tool antiSMASH.
Although the findings and results of the manuscript are interesting, I think some issues need to be addressed.
Basic reporting:
The manuscript is written in a clear and professional language and necessary introduction and background is given.
The structure conforms to PeerJ standards. However, some improvements need to be made. Overall the manuscript organization could be improved. Results that are mentioned in the discussion were not described in the results section. The discussion is already separated into different paragraphs. A headline for each paragraph would make it easier for the reader to follow the discussion.
The main findings (shown as figures) are poorly described in the results section, while the discussion contains a lot of speculation.
The figures are relevant and raw data seem to be supplied, although no sequences were available under the respective accession numbers (still confidential?).

Experimental design

The research is within the scope of the journal, and the research questions are well defined and meaningful.
The method section needs to be defined with more detail:
For example, what were the criteria to define if BGCs in different genomes were of the same cluster type? Blast similarity? Cluster architecture, or domain architecture of modular PKS or NRPS genes?

Validity of the findings

• The main finding presented in the title, is that only one sfp-type phosphopantetheinyl transferase is used for primary and secondary metabolism. However, method and result section are very short and need more detailed explanation. How can you rule out, that another type of PPTase, not detected by your analysis, is used. How similar are the ACPS- and Sfp-like PPTase genes on the nucleotide and protein level?
I can understand that a knock-out mutant might not be possible, since disruption of the primary metabolism is most likely lethal. However, more bioinformatic or in vitro analysis are necessary to really support this. The phylogenetic tree of the PPTase for example just shows that all PPTases in Legionella are related, which is not surprising. More context is needed here. Would it be likely that a third type has been missed by genetic analysis?
• A second major result is the diversity of secondary metabolism in these 15 Legionella strains. However, it is not mentioned what criteria are used to define gene cluster families or how gene clusters split on different contigs were identified.
• Furthermore, it is necessary to put your findings into context with current literature. A short literature search for secondary metabolites in Legionella shows a paper from the Magarvey group analyzing 34 Legionella strains for their secondary metabolism http://www.sciencedirect.com/science/article/pii/S2405805X15300193. How do your results fit in with their findings?

• 28 more Legionella genome sequences are availiable, please comment on why they were not included in the study.

• Lines 272ff: The described methods are all included in the antiSMASH output, but the antiSMASH predictions are not representing actual molecules, but only core structures. How were the circular molecules (Figure 6) predicted? Were any tailoring enzymes taken into account for such predictions? I would suggest to move such predicted structures to the supplementary material, and to emphasize the theoretical nature of the proposed structures, so that readers not familiar with these methods do not confuse them for real structures.
• Line 305 - 306: Please specify what you mean by “Additionally, several other conserved clusters appear to have a similar genetic organization. “

• Line 308ff: The phrasing of this paragraph seems confused.
• Figure 2: Which cluster type is IsnAB? Clusters O-U are not found in Figure 1 or 6.
• Figure 4 & 5: It should be highlighted which functional domain belongs to which gene or the size of the functional domains should be reduced, so that it fits the respective gene size.
• Figure 4 & 6: Why is the same structure predicted for two clusters with a completely different set of genes? (G/I) Why is the polyketide part of the hybrid cluster (I) not taken into account for structure prediction?

Additional comments

No comments.

---

## Round 0.2 · Minor Revisions

· Academic Editor

Minor Revisions

Please address the remaining concerns, especially regarding the strength of your conclusions over the ACP-modifying ability of your Sfp. You may also want to move Supp. Figures 3 and 5 to the main text . The legend to fig. 2 mentions "Figures 1 & 5.", which I believe you to mean "Figures 1 and Supp. Fig. 5"

Reviewer 1 ·

Basic reporting

The manuscript's clarity and focus on what it has ; identification of secondary metabolism pathways and products have improved significantly however there some serious flaws that result from an attempt to extend the conclusions without a relevant, direct evidence. These will require a minor but critical revision so that the MS will reflect its results and the main strength which is the prediction of secondary metabolites

Experimental design

Methods section
PPTase in vitro activity testing
This section describes the cloning and expression of L par PPTase but does not describe purification and the measurements of activity of this PPtase on its substrates that are described in the results section indigoidine and legioliulin
The authors should add those in the main text or change the subheading
Results:
At this point there is no need to use full description of the PPTase name as this should have been described in the introduction section above
Only a single PPTase was identified, which activates natural product biosynthesis in vitro
The authors use indigoidine (blue pigment) production in E coli following coexpression of LndC (encoding an NRP that synthesize indigoidine) and Sfp –like PPTase of legionella. This experiment proves that LparPPTsae recognize and activates NRPS. This is an expected result for Sfp like PPTase and has nothing to do with a single PPTase , a term related for Sfp lke PPTase that takes over the role of AcpS and activates ACP , which is not shown here and is merely relies on non identification of an AcpS . Thus the authors can use this experiment as a clue for having an expected activity of sfp like PPTase to a known NRPS system but not beyond it.
The fact that E coli EntD, its PPTase for secondary metabolism does not activate LndC or other NRPSes like BPSA that make indigoidine was already shown in other studies and has never been used as an evidence for more than a clue for expression of a broad spectrum sfp like PPTase.
Inhibition of legioliunin production and growth in L.parisiensis
The importance and significance of this experiment is that it correlates LParPPTase inhibition to antibacterial activity ( killing of L. pariesiensis), this result now Supplement 3 is relevant enough to be moved to the main text.
However, again a proof of a single Sfp like PPTase is not drawn here as the existence of two PPTase for l secondary metabolism has never been shown, and the proof of activity of this PPTase in fatty acid synthesis ie upon ACP is missing

Validity of the findings

The manuscript's clarity and focus on what it has ; identification of secondary metabolism pathways and products have improved significantly however there some serious flaws that result from an attempt to extend the conclusions without a relevant, direct evidence. These will require a minor but critical revision so that the MS will reflect its results and the main strength which is the prediction of secondary metabolites
The dependence of several enzymes for secondary metabolites synthesis (polyketides and non ribosomal peptides) on activation of their carrier proteins on a single Sfp type PPTase, is a canonic and well established fact in PPTase classification. A second PPTase , which activates only ACP for fatty acid synthesis, a primary metabolism is designated AcpS. Most bacterial species posses these two.
Thus the term a single PPTase implies PPTase that activate both primary, fatty acid synthase, and secondary metabolism polyketide synthases and non ribosomal peptide synthases.
The authors wrongly make an extension of the term " single" PPTase to the field of secondary metabolites, for which there is no known another," third" PPTase , thus finding that all the secondary metabolism in legionella is mediated by a single Sfp like PPTase, is in accord with the current knowledge and has nothing new or notable.
Therefore, the activity of this broad spectrum legionella PPTase towards ACP is not shown as was suggested in former review ( can be easily accomplished) therefore the correct main point of this MS is: "identification and prediction secondary metabolism dependent on a broad spectrum SFP type" but single is not shown and proven beyond absence in genomic analysis
In the absence of a direct proof for activation of ACP the authors should be more conservative Re the conclusion that a single PPTase activates fatty acid synthesis and should not misleadingly use the term "single" PPTase for secondary metabolism functions that are already expected to be mediated by a single Sfp like PPTase.
see

Additional comments

Discussion
The authors inaccurately use an analogy to Pseudomonas aeroginosa which also posses a single PPTase. For PA species the term "single" PPtase was correctly assigned following a direct proof that it takes over both primary metabolism and secondary metabolism functions , which is clearly not the case here. The comparison to PA if anything just highlights this flaw in assigning single PPTase correctly.

·

Basic reporting

The submission adhers to the PeerJ policies, just as before.

The Legionella parisiensis DSM 19216 genome is now available from NCBI as well.

Experimental design

The new legend of figure 2 explains what confused me before, thanks.

Validity of the findings

I agree with response from the authors about the possibility of a new unknown kind of PPTase being involved.

Additional comments

Thank you for attempting to obtain the Celera Assembler parameters, I understand
that external sequencing providers might not keep the documentation around.
Interestingly, the assembly notes on LSOG01000000 state that it's an Illumina GAII
assembly done with Newbler. Possibly that slipped in during the genome submission process.
This metadata inconsistency does in no way affect this manuscript, though.

I understand the decision of the authors to not include LN614827, CP011105 or LN681225 for
the comparison.

Minor issues:
* Some of the references seem to be duplicated in the resubmission. A cursory glance over the reference list
shows Lechner et al. 2011, and Weber et al. 2013 to be duplicated.

Reviewer 3 ·

Basic reporting

Structure and clarity have much improved.

Experimental design

All standards have been met.

Validity of the findings

All standards have been met.

Additional comments

The manuscript is much clearer now, all issues have been addressed adequately.

---

## Round 0.3 · accepted · Accept

· Academic Editor

Accept

You have successfully adressed all of our reviewer's concerns. I have meanwhile noticed that there is a wrong reference in the paper:

"This PPTase may therefore be capable of activating all different carrier proteins involved in polyketide and non-ribosomal peptide biosynthesis as well as fatty acid biosynthesis, a part of the primary metabolism as has been seen before (Losick & Isberg, 2006)."

This reference ("NF-κB translocation prevents host cell death after low-dose challenge by Legionella pneumophila" Journal of Experimental Medicine , 203 , 2177-2189 ) does not seem, at first glance, to relate to phosphopantetheinyl transferases, and may have been introduced by mistake by your reference managing software. Please address this with our production staff.

Reviewer 1 ·

Basic reporting

OK now

Experimental design

OK

Validity of the findings

OK

Additional comments

none